# In Vitro and In Vivo Virulence Study of *Listeria monocytogenes* Isolated from the Andalusian Outbreak in 2019

**DOI:** 10.3390/tropicalmed8010058

**Published:** 2023-01-12

**Authors:** Andrea Vila Domínguez, Marta Carretero Ledesma, Carmen Infante Domínguez, José Miguel Cisneros, Jose A. Lepe, Younes Smani

**Affiliations:** 1Institute of Biomedicine of Seville, IBiS, University Hospital Virgen del Rocio/CSIC/University of Seville, 41013 Sevilla, Spain; 2Clinical Unit of Infectious Diseases, Microbiology and Parasitology, University Hospital Virgen del Rocio/CSIC/University of Seville, 41013 Sevilla, Spain; 3CIBER de Enfermedades Infecciosas (CIBERINFEC), Instituto de Salud Carlos III, 28222 Madrid, Spain; 4Department of Molecular Biology and Biochemical Engineering, Andalusian Center of Developmental Biology, CSIC, University of Pablo de Olavide, 41013 Seville, Spain

**Keywords:** *Listeria*, outbreak, animal model, virulence, pathogenicity

## Abstract

In 2019, the biggest listeriosis outbreak by *Listeria monocytogenes* (Lm) in the South of Spain was reported, resulting in the death of three patients from 207 confirmed cases. One strain, belonging to clonal complex 388 (Lm CC388), has been isolated. We aimed to determine the Lm CC388 virulence in comparison with other highly virulent clones such as Lm CC1 and Lm CC4, in vitro and in vivo. Four *L. monocytogenes* strains (Lm CC388, Lm CC1, Lm CC4 and ATCC 19115) were used. Attachment to human lung epithelial cells (A549 cells) by these strains was characterized by adherence and invasion assays. Their cytotoxicities to A549 cells were evaluated by determining the cells viability. Their hemolysis activity was determined also. A murine intravenous infection model using these was performed to determine the concentration of bacteria in tissues and blood. Lm CC388 interaction with A549 cells is non-significantly higher than that of ATCC 19115 and Lm CC1, and lower than that of Lm CC4. Lm CC388 cytotoxicity is higher than that of ATCC 19115 and Lm CC1, and lower than that of Lm CC4. Moreover, Lm CC388 hemolysis activity is lower than that of the Lm CC4 strain, and higher than that of Lm CC1. Finally, in the murine intravenous infection model by Lm CC388, higher bacterial loads in tissues and at similar levels of Lm CC4 were observed. Although a lower rate of mortality of patients during the listeriosis outbreak in Spain in 2019 has been reported, the Lm CC388 strain has shown a greater or similar pathogenicity level in vitro and in an animal model, like Lm CC1 and Lm CC4.

## 1. Introduction

Listeriosis is a zoonosis related to the consumption of contaminated foods and is classified as one of the most serious foodborne diseases [1]. *Listeria monocytogenes* causes human listeriosis in well-defined risk groups (people older than 65 years, people with immunosuppression, women who are pregnant and neonates) and can evolve into invasive disease. Clinical presentation includes non-invasive syndromes such as febrile gastroenteritis, and invasive syndromes, mainly sepsis and meningoencephalitis, as well as focal infections such as pneumonia, endocarditis and septic arthritis [2].

In 2018–2019, listeriosis presented a high lethality rate across the European Union, reaching 8.9%, with many listeriosis cases associated with epidemic outbreaks (frozen corn products in Hungary and fish products in Denmark, Germany and France) [3]. The number of outbreaks in 2019 was 21, which is 50% higher than that in 2018 [3].

While all *L. monocytogenes* isolates are clinically reported to be invariably virulent, this pathogen is genetically highly heterogeneous regarding its pathogenesis [2]. It is clustered into four lineages and four serogroups. Several isolates are from lineage (I)/serotypes (1/2b, 3b and 4b), and from lineage (II)/serotypes (1/2a, 1/2c, 3a and 3c) [4]. It is well-known that serotype (4b) is associated with the most listeriosis in humans, and serotypes (1/2a, 1/2b, 1/2c and 4b) are highly expressed in clinical isolates and food [5]. Only a few worldwide disseminated clonal groups, defined by MLST or next-generation sequence, caused the many outbreaks of listeriosis [6]. Clonal complexes (CCs) CC1, CC2, CC4 and CC6 are the complexes most commonly associated with clinical isolates causing listeriosis [2].

The pathogenesis of *L. monocytogenes* is mediated by components present in their surfaces or released in the extracellular medium. The main virulence factors are Internalin A (InlA), Internalin B (InlB) and Internalin F (InlF) [1,7]. These proteins interact with the host cell E-cadherin via an intracellular route and help *L. monocytogenes* cross the blood–brain barrier (in the case of InlF) [7,8]. Further external and internal proteins have been reported to be involved in the pathogenesis of *L. monocytogenes* and have been characterized [1,9,10]. For numerous types of proteins, such as listeriolysin, hemolysin, phospholipase C and ActA (actin-based intracellular motility), mutants depleted in these proteins presented lower virulence [10].

Adhesion is a priority step in *L. monocytogenes* infections which allow *L. monocytogenes* to enter eukaryotic cells such as the epithelial cells of the lungs and intestine [11]. They invade epithelial cells by endocytosis, which are trapped by a vacuole or phagosome; then, they spread out throughout the cytoplasm, after disrupting phagosomes, by secreting listeriolysin [10,11,12].

In 2019, the biggest listeriosis outbreak in the Andalusian region was reported [13], with 207 confirmed cases, resulting in the hospitalization of 141 patients and the death of 3 patients, resulting in a mortality rate of 1.45% [13]. In this outbreak, a ST388 (4b serotype; CC388) strain was isolated in meat and retail products [13].

Here, we determined the virulence of the *L. monocytogenes* isolate that caused the listeriosis outbreak in Spain (Lm CC388) and associated it with the low mortality rate of patients. Thus, we compared the virulence of this strain with that of other highly virulent strains such as Lm CC1 and Lm CC4, in an in vitro and in vivo model of infection.

## 2. Material and Methods

### 2.1. Bacteria and Conditions of Bacterial Growth

*L. monocytogenes* from 4b serotype were used in this study: the strain causing the Andalusian listeriosis outbreak in 2019 belonging to CC338 (Lm CC388); two human hypervirulent strains belonging to CC1 (Lm CC1) and CC4 (Lm CC4); and a reference strain ATCC 19115, belonging to CC2 (LGC Standards, London, UK).

The strains were cultured in Mueller Hinton broth (MHB) at 37 °C, with an incubation time of 24 h. For experiments of A549 cells culture, the bacterial inoculum was first washed with PBS and then resuspended in DMEM.

### 2.2. Human Cell Culture

The epithelial cell line A549 culture at 5% CO_2_ and 37 °C was performed as we previously described [14]. We grew A549 cells in DMEM in the presence of HEPES (1%), fetal bovine serum (10%) and antibiotics (amphotericin B, gentamicin, and vancomycin) (Invitrogen, Spain). For infection experiments, the A549 cells were first washed with PBS and then incubated in non-completed DMEM (free of fetal bovine serum, amphotericin B, gentamicin and vancomycin).

### 2.3. Adhesion and Internalization Assays

The A549 cells were infected with the ATCC 19115, Lm CC1, Lm CC388 and Lm CC4 strains at MOI of 50 for 2 h at 5% CO_2_ and 37 °C, as previously described [14]. After washing the infected A549 cells with phosphate buffered saline, they were incubated with Triton X-100 (0.5%) for lysis. Then, the lysates were diluted and spread in plates of blood agar plates (ThermoFisher, Waltham, MA, USA) and incubated for 24 h at 37 °C, and the number of colonies that attached to A549 cells was determined.

In addition, to determine the number of colonies that entered inside the A549 cells, the wells were washed with phosphate buffered saline and incubated for 30 min in the presence of DMEM plus gentamicin (256 µg/mL), in order to kill the bacteria present in the area. Then, the wells were washed with phosphate buffered saline to remove gentamicin. The number of colonies that entered inside A549 cells was determined as described above.

### 2.4. Cellular Viability

The eukaryotic cells were infected with the ATCC 19115, Lm CC1, Lm CC388 and Lm CC4 strains at a MOI of 50 for 24 h, and incubated with MTT for 4 h as we previously described [15]. The % of cellular viability was determined from the optical density at 550 nm.

### 2.5. Hemolytic Activity

The hemolytic activity in the murine erythrocytes after incubation with the ATCC 19115, Lm CC1, Lm CC388 and Lm CC4 strains for 4 h was determined. Blood collected from the retro-orbital sinus area in mice was incubated in glass spheres while stirring for 20 min to eliminate any sera and to free the hemoglobin. Fifty microliters of the supernatant with erythrocytes was incubated with an equal volume of 1 × 10^8^ CFU/mL of each bacterial strain for 4 h at 37 °C and centrifuged for 3 min at 3000× *g*. Ninety microliters of the supernatant relocated into a 96-microwell plate was used to determine the absorbance of released hemoglobin at 550 nm (BioRad 680 plate reader, Portuguese, Spain). Triton X-100 at 0.1% was used as a positive control to obtain full hemolysis.

### 2.6. Murine Model of Infection

An infection model of BALB/c female mice (18–20 g) (Charles River, Barcelona, Spain) with the four *L. monocytogenes* strains was performed by intravenous bacterial administration [16]. All animals were anaesthetized with sodium thiopental (B. Braun Medical S.A., Barcelona, Spain) to minimize suffering. Groups of six animals for each strain were administered 0.5 mL of different bacterial inoculum, and their survival was monitored for 7 days. The 0% lethal dose (LD_0_), 50% lethal dose (LD_50_) and 100% minimum lethal dose (MLD_100_) values were determined [14]. To determine the bacterial concentrations in blood (CFU/mL) and tissues (CFU/g), groups of five animals were administered 5 log CFU/mL of each strain intravenously. They were killed by an intraperitoneal administration of sodium thiopental 48 h post-bacterial administration. Spleens, lungs, kidneys and livers were removed and homogenized in 2 mL of NaCl 9% by a Stomacher 80 homogenizer (Tekmar Co., Cincinnati, OH, USA).

### 2.7. Statistical Analysis

The data are represented as means ± SEMs. For the in vitro and in vivo experiments, Student’s *t*-tests and ANOVAs and post hoc Dunnett’s were used, respectively (IBM SPSS Statistics 22 software). *p* values < 0.05 were considered for statistical difference.

## 3. Results

### 3.1. L. monocytogenes Adherence and Invasion in Host Cells

We determined whether the Lm CC388 strain presented a similar interaction with A549 cells to those of ATCC 19115, Lm CC1 and Lm CC4 strains.

Figure 1A showed that the adherence of the Lm CC388 strain to A549 cells was higher than the adherence of the ATCC 19115 strain and the Lm CC1 strain (0.8 × 10^6^ CFU/mL versus 0.1 × 10^6^ and 0.6 × 10^6^ CFU/mL), but lower than that of the Lm CC4 strain (0.8 × 10^6^ CFU/mL versus 1.5 × 10^6^ CFU/mL).

Moreover, the Lm CC388 strain counts inside A549 cells were higher than those in the ATCC 19115 strain and the Lm CC1 strain (1.5 × 10^4^ CFU/mL versus 1.1 × 10^4^ and 1.4 × 10^4^ CFU/mL), but lower than in the Lm CC4 strain (1.5 × 10^4^ CFU/mL versus 2.8 × 10^4^ CFU/mL). These data indicate that the host cells interact with the Lm CC388 strain less than with the Lm CC4 strain (Figure 1B).

### 3.2. Cytotoxicity Activity of L. monocytogenes

We also determined whether the Lm CC388 strain presented similar cytotoxicity to the ATCC 19115, Lm CC1 and Lm CC4 strains. The assay of cell viability showed that ATCC 17978, Lm CC1 and Lm CC388 strains reduced the cell viability to 75.4%, 66.7% and 49.1%, respectively. In contrast, the Lm CC4 strain reduced the cell viability to 48% (Figure 2). These results show that the cytotoxicity of the Lm CC388 strain is lower than that of the Lm CC4 strain.

### 3.3. Hemolytic Activity of L. monocytogenes

Additionally, we determined the hemolytic activity of Lm CC388 versus other *L. monocytogenes* strains. The hemolytic assay showed that 4 h of incubation in the erythrocytes with the ATCC 17978, Lm CC1 and Lm CC388 strains increased hemolysis to 1.7 × 10^−1^, 1.5 × 10^−1^ and 1.8 × 10^−1^ absorbance at 550 nm, respectively. However, the Lm CC4 strain increased the hemolysis to 2.4 × 10^−1^ absorbance at 550 nm (Figure 3). These results showed that the hemolytic activity of the Lm CC388 strain is lower than that of the Lm CC4 strain.

### 3.4. Pathogenesis of L. monocytogenes in an Intravenous Murine Infection Model

To analyze the virulence of the Lm CC388 strain and other *L. monocytogenes* strains, an intravenous murine infection model was performed. The mortality rates of mice were concentration-dependent on bacteria for the ATCC 19115, Lm CC1, Lm CC388 and Lm CC4 strains (Table 1). LD_0_, LD_50_ and MLD_100_ for the Lm CC388 and Lm CC4 strains were lower than those for the ATCC 19115 strain: 3, 3.5 and 5 log CFU/mL versus 5, 5.5 and 6 log CFU/mL, respectively, with ratios of 0.6, 0.65 and 0.83, respectively. LD_50_ for the Lm CC388 and Lm CC4 strains was lower than that for the Lm CC1 strain: 3.53 log CFU/mL versus 4.11 log CFU/mL (Table 1).

In addition, we performed the same murine model to determine the dissemination of the Lm ATCC 19115, Lm CC1, CC388 and Lm CC4 strains to different tissues, administering bacterium to mice intravenously at 5 log CFU/mL. The removed tissues and blood showed higher bacterial loads in animals with the Lm CC388, Lm CC1 and Lm CC4 strains than in animals with the ATCC 19115 strain (Figure 4). We observed differences between the Lm CC388 strain, and the Lm CC4, Lm CC1 and ATCC 19115 strains in the spleen (9.5 log CFU/g versus 9.5, 8.1 and 5.7 log CFU/g), lungs (6.1 log CFU/g versus 5.4, 3.9 and 2.3 log CFU/g), liver (8.1 log CFU/g versus 7, 7.2 and 5.1 log CFU/g), kidneys (5.4 log CFU/g versus 4.7, 3.9 and 1.1 log CFU/g) and blood (3.9 log CFU/mL versus 3.1, 2.6 and 0.5 log CFU/mL). These data indicated that the Lm CC388 strain showed similar dissemination and infective capacity to that of the Lm CC4 strain in vivo.

## 4. Discussion

The present study provides new results upon investigating the virulence properties of the isolate causing the listeriosis outbreak (Lm CC388) in southern Spain in 2019. Here, we provide the first evidence of the degree of virulence of this isolate in comparison with other hypervirulent isolates included in the 4b serotype, in vitro and in vivo.

This study showed that no significant differences exist between the virulence of the Lm CC388 strain and the virulence of other hypervirulent strains such as Lm CC1 and Lm CC4. Previous independent work showed that *L. monocytogenes* strains belonging to CC4 are considered hypervirulent [17,18,19,20]. However, to date, the question is whether the strain causing the listeriosis outbreak in the Andalusian region presents a similar virulence to strains belonging to CC1 and CC4.

Here, we showed that the Lm CC388 strain presents more interactions with the host cells, causes more hemolysis and is more pathogenic in an animal model than the Lm CC1 and ATCC 19115 strains, suggesting that the virulence of the Lm CC388 strain has contributed to the development of listeriosis in Spain. Evidence supports this hypothesis because different studies have reported that strains belonging to CC1 are major contributors to human listeriosis, and are considered hypervirulent strains as well [4,19,21,22,23] as they cause further invasion and colonization of tissue, such as the intestinal lumen [17]. Infections caused by these strains are linked to an unfavorable evolution of listeriosis, triggering rhombencephalitis and miscarriage [18].

Noteworthily, some hypovirulent strains belonging to CC2, CC8, CC9 and CC14 and associated with the natural environment are also involved in listeriosis in individuals who are immunocompromised [18,19,22,24]. However, hypervirulent strains causing listeriosis have been well-reported to be commonly found in clinical environments [4].

To induce an infection, *L. monocytogenes* expresses virulence genes organized into groups such as the Internalin gene operon (encoded by *inlA* to *inlK*), listerial pathogenicity islands (encoded by *LIPI*-1to *LIPI-4*) and stress survival islets (SSIs) [4]. Accordingly, two recent studies showed that strains belonging to CC388 harbor LIPI-4, but not inLF, inlG, SSI-1 and SSI-2 [25,26]. This pathogenic island is also present in isolates belonging to CC4 that cause cerebral and placental tissue tropism [17,20]. More genome sequencing studies are required to identify the genes involved in the virulence of the Lm CC388 strain and to match them with those of the Lm CC1 and Lm CC4 strains.

Even though the Lm CC388 strain has shown levels of virulence greater than or similar to those of the Lm CC1 and Lm CC4 strains, a lower mortality rate (1.38%) was reported in patients infected by Lm CC388 during the biggest listeriosis outbreak in Spain in 2019 [13]. The low mortality in this outbreak contrasts with that reported in the listeriosis outbreaks in the USA, Denmark and South Africa, which had higher mortality rates (from 22 to 41%) [27,28,29].

The results of this study suggest that this positive discrepancy in mortality is not due to a lower virulence of the isolate causing the listeriosis outbreak. Ongoing epidemiological and clinical studies of the outbreak might help to understand these large prognostic differences.

## 5. Conclusions

Although a lower rate of mortality of patients during the listeriosis outbreak in Spain in 2019 has been reported, the Lm CC388 has shown a greater or similar pathogenicity level in vitro and in animal models like Lm CC1 and Lm CC4.

## Figures and Tables

**Figure 1 tropicalmed-08-00058-f001:**
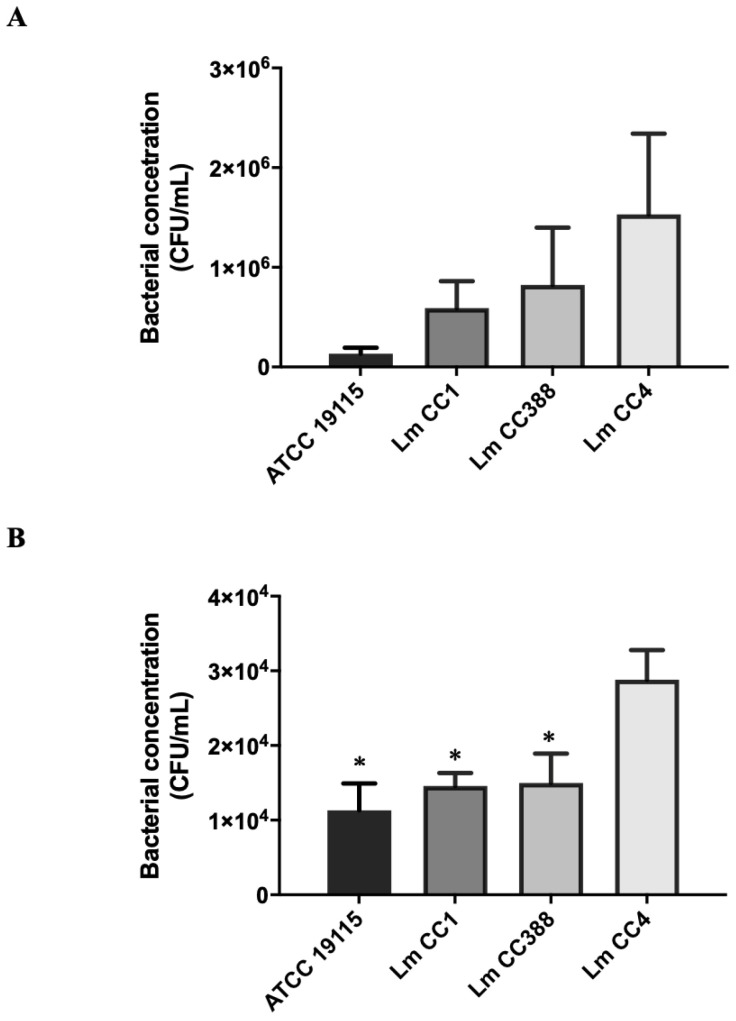
*L. monocytogenes* adherence and invasion in host cells. (**A**) Analysis of bacterial adherence in A549 cells incubated with *L. monocytogenes* Lm CC88, Lm CC1, Lm CC4 and ATCC 19115 strains. (**B**) Analysis of bacterial invasion in A549 cell incubated with *L. monocytogenes* Lm CC88, Lm CC1, Lm CC4 and ATCC 19115 strains. Data are the means of 3 repetitive assays. *: *p* < 0.05 versus Lm CC4 strain.

**Figure 2 tropicalmed-08-00058-f002:**
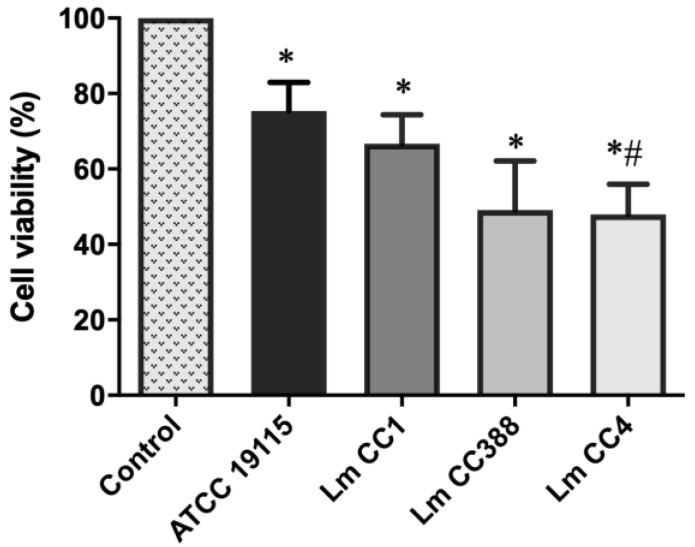
Cytotoxic activity of *L. monocytogenes*. Cell viability of A549 cells with *L. monocytogenes* Lm CC388, Lm CC1, Lm CC4 and ATCC 19115 strains for 24 h. Data are the means of 3 repetitive assays. *: *p* < 0.05 versus control, #: *p* < 0.05 versus ATCC 19115.

**Figure 3 tropicalmed-08-00058-f003:**
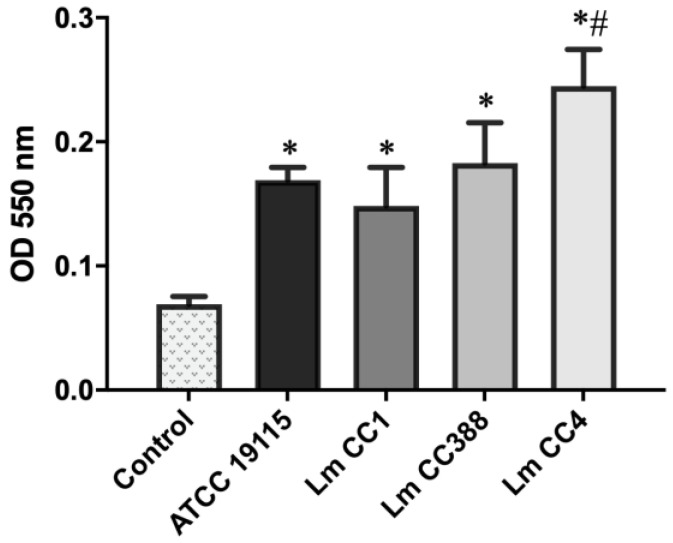
Hemolytic activity of *L. monocytogenes*. Hemolysis of murine erythrocytes after 4 h of incubation with *L. monocytogenes* Lm CC388, Lm CC1, Lm CC4 and ATCC 19115 strains. Data are the means of 3 different assays. *: *p* < 0.05 versus control, #: *p* < 0.05 versus ATCC 19115 or Lm CC1.

**Figure 4 tropicalmed-08-00058-f004:**
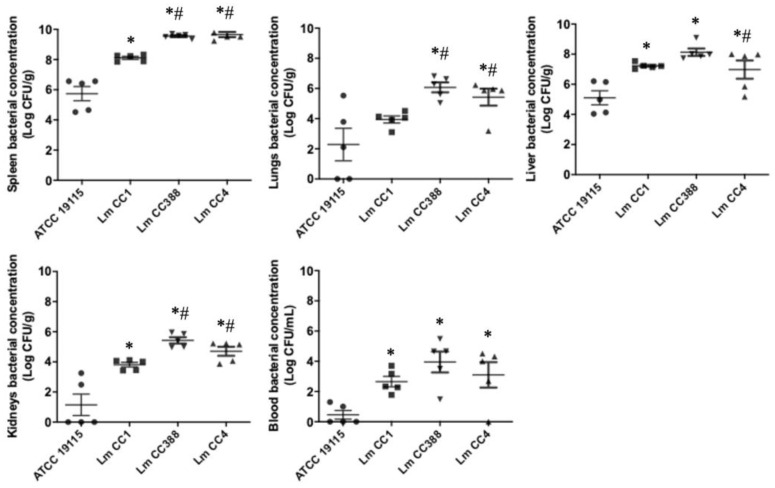
Bacterial burden in intravenous murine model by *L. monocytogenes*. Bacterial concentrations in spleen, lungs, liver, kidneys and in the murine model of peritoneal sepsis by *L. monocytogenes* Lm CC388, Lm CC1, Lm CC4 and ATCC 19115 strains at 5 log_10_ CFU/mL. Data are represented as mean ± SEM. *: *p* < 0.05 versus ATCC 19115, #: *p* < 0.05 versus Lm CC1.

**Table 1 tropicalmed-08-00058-t001:** Seven days of mortality monitoring in mice inoculated intravenously with *L. monocytogenes*.

	LD_0_ (log CFU/mL)	LD_50_ (log CFU/mL)	MLD_100_ (log CFU/mL)
ATCC 19115 (CC2)	5	5.5	6
CC1	3	4.1	5
CC388	3	3.5	5
CC4	3	3.5	5

LD: lethal dose; MLD_100_: minimal lethal dose 100.

## Data Availability

Not applicable.

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
