# Peer review of "In Vitro and In Vivo Virulence Study of Listeria monocytogenes Isolated from the Andalusian Outbreak in 2019"

_tropicalmed, 2023, doi:10.3390/tropicalmed8010058_

Round 1

Reviewer 1 Report

The presented manuscript discusses an important issue, which is the characterization of L. monocytogenes, especially those responsible for the outbreak.

Some comments:

- add information about non-invasive listeriosis to the introduction;

- give examples of the most important listeriosis epidemics in recent years;

- the authors reported that the most important virulence factors are internalins. However, be aware of LLO and ActA. Please add some information about LIPI-1 to the introduction.

Author Response

Thank you very much for your comments.
-Information on non-invasive listeriosis as well as examples of the others important listeriosis epidemics in Europe is included in the Introduction section (lines 40-46). 
-Information about LIPI-1 included in the Discussion section (lines 912-913).

Reviewer 2 Report

The study provides interesting data about the virulence characteristic of the listeriosis-causing isolates. These findings can be of interest to readers.

Author Response

I appreciate your comment, but he manuscript was submitted to MDPI for editing in English and was edited in November 2022 (English edition ID: English-52573).